# Latency reversal plus natural killer cells diminish HIV reservoir in vivo

Jocelyn T. Kim [1,11✉], Tian-Hao Zhang[2,11], Camille Carmona [3], Bryanna Lee[1], Christopher S. Seet [4], Matthew Kostelny[3], Nisarg Shah[3], Hongying Chen[3], Kylie Farrell[3], Mohamed S. A. Soliman [3], Melanie Dimapasoc [3], Michelle Sinani[1], Kenia Yazmin Reyna Blanco[1], David Bojorquez [3], Hong Jiang[5], Yuan Shi [5], Yushen Du[5], Natalia L. Komarova[6], Dominik Wodarz [7], Paul A. Wender[8], Matthew D. Marsden[9], Ren Sun[5,10] & Jerome A. Zack [3,4]

HIV is difficult to eradicate due to the persistence of a long-lived reservoir of latently infected cells. Previous studies have shown that natural killer cells are important to inhibiting HIV infection, but it is unclear whether the administration of natural killer cells can reduce rebound viremia when anti-retroviral therapy is discontinued. Here we show the administration of allogeneic human peripheral blood natural killer cells delays viral rebound following interruption of anti-retroviral therapy in humanized mice infected with HIV-1. Utilizing genetically barcoded virus technology, we show these natural killer cells efficiently reduced viral clones rebounding from latency. Moreover, a kick and kill strategy comprised of the protein kinase C modulator and latency reversing agent SUW133 and allogeneic human peripheral blood natural killer cells during anti-retroviral therapy eliminated the viral reservoir in a subset of mice. Therefore, combinations utilizing latency reversal agents with targeted cellular killing agents may be an effective approach to eradicating the viral reservoir.

[1] Department of Medicine, Division of Infectious Diseases, University of California Los Angeles, Los Angeles, CA 90095, USA. [2] Molecular Biology Institute, University of California Los Angeles, Los Angeles, CA 90095, USA. [3] Department of Microbiology, Immunology, and Molecular Genetics, University of California Los Angeles, Los Angeles, CA 90095, USA. [4] Department of Medicine, Division of Hematology and Oncology, University of California Los Angeles, Los Angeles, CA 90095, USA. [5] Department of Molecular and Medical Pharmacology, University of California, Los Angeles, CA 90095, USA. [6] Department of Mathematics, University of California, Irvine, Irvine, CA 92697, USA. [7] Department of Population Health and Disease Prevention, Program in Public Health Susan and Henry Samueli College of Health Sciences, University of California, Irvine, Irvine, CA 92697, USA. [8] Department of Chemistry and Department of Chemical and Systems Biology, Stanford University, Stanford, CA 94305, USA. [9] Department of Microbiology and Molecular Genetics and Department of Medicine, Division of Infectious Diseases, School of Medicine, University of California, Irvine, Irvine, CA 92697, USA. [10] School of Biomedical Sciences, LKS Faculty of Medicine, The University of Hong Kong, Hong Kong, China. [11] These authors contributed equally: Jocelyn T. Kim, Tian-Hao Zhang. ✉email: jocelynkim@mednet.ucla.edu

Even though antiretroviral therapy (ART) can effectively halt HIV replication, ART must be maintained for life because latent HIV-1 infected cells persist and initiate active virus replication if ART is discontinued[1–3]. Experimental approaches for eliminating the latent reservoir have included myeloablation followed by transplantation of cells lacking co-receptors for virus infection, genome editing of the latent provirus, or use of viral inducers in the presence of ART (reviewed in ref. [4]). Various latency reversing agents (LRAs) have been used to kick (or shock) infected cells out of latency and induce their death by virus-induced cytopathic effects, but a kick alone may not be sufficient in eliminating the HIV reservoir[5]. Indeed, we have previously described a synthetic bryostatin analog SUW133, which by itself reversed latency and induced the death of a subset of previously latent cells[6–8], suggesting that the addition of a dedicated kill agent could further diminish the viral reservoir. Kill approaches have included immunological therapies that target HIV-infected cells by enhancing endogenous antiviral immune responses or harnessing of antibody-based or effector cell therapies (reviewed in[9]). One combinatorial kick and kill approach comprised of multiple LRAs combined with broadly neutralizing antibodies (bNAbs) 3BNC117, 10-1074, and PG16 administered during ART decreased viral rebound after ART interruption in humanized mice infected with HIV-1[10]. Other kick and kill approaches utilizing a TLR7 agonist and Ad26/MVA vaccine[11] or bNAb PGT121[12], decreased viral rebound after ART interruption in rhesus monkeys infected with SIV or SHIV, respectively. These preclinical studies demonstrate the enhancement of endogenous antiviral immunity or the addition of bNAbs as a dedicated kill agent to LRAs diminishes viral rebound from the reservoir. Other potential strategies towards an HIV cure, including the combination of latency reversal with anti-HIV cellular therapies, represent promising approaches that are currently understudied. Here we show injections of allogeneic human peripheral blood natural killer (NK) cells alone slow and sometimes prevent a viral rebound in HIV-infected humanized mice. We also demonstrate a single administration of the protein kinase C modulator and latency reversing agent SUW133 followed by injections of these NK cells during ART further decreases rebound frequency and delays rebound when it occurs, which provides proof-of-concept that a kick and kill strategy can effectively target the HIV reservoir. Importantly, we utilize genetic barcoded virus technology to show these treatments also reduce the diversity of the HIV reservoir, thus demonstrating adjunct interventions may eliminate the reservoir towards a complete functional cure.

## Results

### NK cells are activated by allogeneic HIV-infected CD4 + T cells.
NK cells rapidly target and kill HIV-infected cells, which is important to early control of HIV infection and AIDS[13–19]. However, endogenous cytotoxic effector cells including NK cells may be diminished or impaired during chronic HIV infection and not fully restored despite achieving viral suppression[20–23]. Allogeneic peripheral blood NK cells have been shown to efficiently inhibit viral replication in vitro[24] and acute HIV infection in a humanized mouse model[25]. Recently, activated NK cells primed with IL-15 inhibited latently infected cells from propagating infection in vitro[26], suggesting NK cells as a promising therapy to treat viral rebound. Thus, we hypothesized that exogenously administered NK cells might be a valuable component of HIV reservoir elimination strategies intended to delay or prevent HIV rebound after ART interruption.

We first sought to investigate whether peripheral blood NK cells could be utilized as an efficient kill agent against cells productively infected with HIV. We obtained human peripheral blood NK cells from the peripheral blood mononuclear cells (PBMCs) of four healthy donors using magnetic bead isolation. We used Uniform Manifold Approximation and Projection (UMAP) to visualize subpopulations among the CD56$^+$CD3$^-$ cells from concatenated and individual samples by flow cytometry (Supplementary Fig. 1a,b). We next manually assigned clusters based on the differential expression of eleven surface markers and identified five subpopulations, including the canonical NK cell subsets: CD56$^{dim}$CD16$^+$ (populations 1-4) and CD56$^{bright}$CD16$^-$ cells (population 5) (Supplementary Fig. 1c). As expected, CD56$^{dim}$CD16$^+$ NK cells were the predominant cell type in the peripheral blood (Supplementary Fig. 1d)[27–29]. Among the CD56$^{dim}$CD16$^+$ NK cells, we identified four subpopulations based on the differential expression of CD57, KIR2DL1/S1/S3/S5, and natural cytotoxicity receptor (NCR) NKp44 (Supplementary Fig. 1c)[29,30]. In comparison to CD56$^{dim}$CD16$^+$ NK cells, the CD56$^{bright}$CD16$^-$ cells demonstrated increased expression of NCR NKp46 and inhibitory receptor NKG2A and decreased expression of KIR2DL1/S1/S3/S5 and CD57, which was consistent with an immature phenotype as previously reported[29–31]. All subpopulations expressed high levels of the activating co-receptors 2B4 and NKp80 and cell stress sensing activating receptor NKG2D and low levels of NCR NKp30[29]. These results were consistent with our current understanding that the peripheral blood contains the classical subsets of CD56$^{dim}$CD16$^+$ NK and CD56$^{bright}$CD16$^-$ cells as well as additional heterogeneous subpopulations.

To assess the function of these NK cells, we isolated allogeneic CD4$^+$ T cells from the PBMCs of healthy donors and performed CD3 and CD28 co-stimulation for three days prior to infection with 800 ng of p24 of X4-tropic NL4-3 or R5-tropic NFNSX for 24 h. NFNSX is derived from NL4-3 with the envelope cloned from the CCR-5 tropic JR-FL[32]. The infected or uninfected CD4$^+$ T cells were washed and cocultured with allogeneic NK cells at an effector-to-target ratio of 1:1 in media containing 20 ng per ml of recombinant human IL-2 (Supplementary Fig. 2a). Cocultures were analyzed 24 and 48 h later to assess activation of NK cells and frequency of live infected T cells by flow cytometry. After 24 h of coculture the allogeneic NK cells exhibited higher levels of IFN-γ production and CD107a degranulation when cocultured with HIV-infected CD4$^+$ T cells compared to allogeneic NK cells cocultured with uninfected CD4$^+$ T cells or NK cells cultured alone (Supplementary Fig. 2b), suggesting that allogeneic NK cells were specifically activated by HIV-infected CD4$^+$ T cells. These results were consistent with previous findings that allogeneic NK cells are efficiently activated by HIV-infected CD4$^+$ T cells[33,34]. In addition, NK cells treated with varying doses of cell-free HIV supernatant in the absence of CD4$^+$ T cells demonstrated no notable increase in levels of intracellular IFN-γ or degranulation (CD107a) after 24 h or intracellular p24 staining after 48 h (Supplementary Fig. 2c, d), suggesting that cell-free virus did not efficiently activate or replicate in NK cells. As expected, the frequency of infected p24$^+$ CD4$^+$ T cells was significantly lower among cocultures of allogeneic NK cells with infected CD4$^+$ T cells compared to cultures containing infected CD4$^+$ T cells without NK cells (Supplementary Fig. 2e, f), indicating that allogeneic NK cells decreased HIV infection in CD4$^+$ T cells.

### NK cells delay viral rebound of R5-tropic HIV after ART interruption.
Next, to study HIV latency in NSG-BLT mice, which achieve a high level of human immune cell reconstitution[35], we intravenously injected NFNSX and monitored acute viremia by qRT-PCR in plasma samples for four weeks. Infected mice were administered ART comprised of

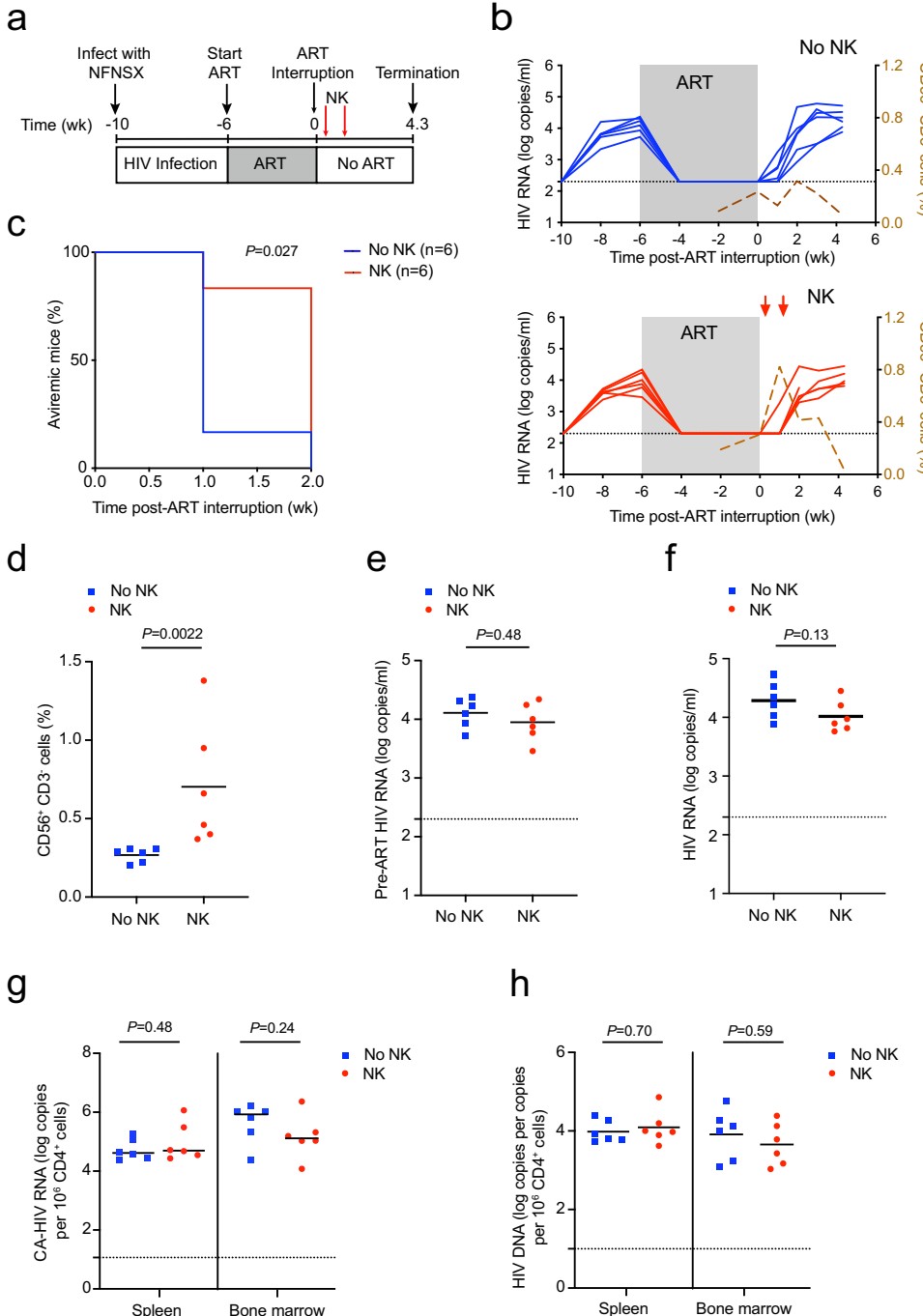

**Fig. 1 NK cells delay time to viral rebound after ART interruption in BLT mice infected with HIV strain NFNSX. a** Schematic representation of experiment; acute infection of TKO-BLT mice with NFNSX for 4 weeks, ART for 6 weeks. One day after ART interruption, $5 \times 10^6$ allogeneic human peripheral blood NK cells were transferred followed by another homologous dose of NK cells 5 days later. The red arrows denote when NK cells were given. **b** Longitudinal plasma viral loads for each infected animal at various timepoints in the No NK (blue) and NK (red) groups. Gray shading indicates an ART treatment period of 6 weeks. A dashed brown line indicates the median frequency of CD56+CD3− cells detected in the blood. **c** Kaplan–Meier curves showing the frequency of aviremic mice after ART interruption. *P* value calculated by log-rank Mantel–Cox test. **d** Frequency of CD56+CD3− cells in the blood 5 days after the first injection of NK cells. **e, f** Plasma viral loads for each animal before ART was initiated (**e**) and at necropsy (**f**). **g, h** Cell-associated "CA" HIV RNA (**g**) and HIV DNA (**h**) levels from the spleen and bone marrow of each infected animal. *n* = 6 biologically independent animals in each group observed over one independent experiment (**b**–**h**). The Black dotted line indicates the detection limit of 2.3 log RNA copies per ml (**b, e, f**) and 1.0 log RNA or DNA copies (**g, h**). Horizontal bars represent the means (**d**–**h**). *P* values were calculated using a two-tailed Mann–Whitney test (**d**–**h**). Source data are provided as a Source Data file.

raltegravir (RAL), emtricitabine (FTC), and tenofovir disoproxil fumarate (TDF) in the animal feed for six weeks (Fig. 1a)[6]. Once viremia was suppressed, ART was interrupted in the presence or absence of $5 \times 10^6$ allogeneic human peripheral blood NK cells, which were administered at one and six days post-ART interruption. The donors for NK cell injections were randomly selected based on donor supply and not chosen based on KIR/HLA genotyping. Injections of NK cells in a mouse cohort were from the same human donor. Because there are no robust virologic or immunologic correlates to definitively identify all HIV reservoir cells, monitoring viral rebound after ART interruption was used as a primary endpoint in vivo. One week after ART interruption, rebound viremia occurred in five out of six (83%) mice in the control group and one out of six (17%) mice in the group receiving NK cells (Fig. 1b). After two weeks of ART interruption, all mice in both groups rebounded. The group receiving NK cells exhibited a delay in viral rebound compared to the control group ($P = 0.027$, Fig. 1c). As expected control mice displayed low frequencies of human CD56$^+$CD3$^-$ cells in the blood due to poor human NK cell development in BLT mice[36] (Fig. 1b, top). Compared to control mice, the mice receiving NK cells had six-fold higher, but overall low levels (~0.8%) of human CD56$^+$CD3$^-$ cells in the blood five days after the administration of NK cells ($P = 0.0022$, Fig. 1d), which then declined thereafter (Fig. 1b, bottom). Indeed, the adoptive transfer of allogeneic NK cells has limited engraftment[37,38]. There were no notable differences between the frequencies of CD56$^+$CD3$^-$ cells in the blood, spleen, and bone marrow at necropsy between the two groups (Supplementary Fig. 3a). Although NK cells can downregulate CD56 expression and subpopulations of long-lived memory-like NK cells can exist[39], our results likely indicate the overall transient survival of adoptively transferred non-engineered NK cells[40,41].

The levels of pre-ART viral infection and human immune cell engraftment likely did not account for differences in viral rebound because notable differences were not observed between the two groups (Fig. 1e and Supplementary Fig. 3b–e). In addition, mice receiving NK cells did not demonstrate diminished frequencies or absolute counts of human CD45$^+$ and CD4$^+$ T cells compared to control mice, suggesting allogeneic NK cells did not indiscriminately kill host immune cells, which is consistent with clinical studies demonstrating the overall safety of allogeneic NK cells in contrast to allogeneic T cells, which induce graft-versus-host disease[42–44].

In addition, we found the level of viremia, cell-associated HIV RNA, and total HIV DNA of the spleens and bone marrow at necropsy were not significantly different between the mice treated with or without NK cells, which was expected given the rebounding virus was allowed to replicate and reseed new target cells during ART interruption (Fig. 1f–h). These results indicate total HIV RNA and DNA measurements several weeks after ART interruption did not capture the dynamic effect that NK cells had on the early viral rebound.

**NK cells also delay rebound of X4-tropic barcoded HIV.** Next, we investigated whether HIV-1 barcoded technology could quantify the effect that allogeneic human peripheral blood NK cells had on rebounding viral clones after ART interruption[8,45]. As such, we recently developed a genetically barcoded HIV-1 containing a 21 bp genetic barcode inserted upstream of a hemagglutinin tag in the nonfunctional vpr region of the HIV strain NL-HA[8,46], which is an X4-tropic near full-length, replication-competent, pathogenic strain of NL4-3 (Fig. 2a and Supplementary Fig. 4a–c).

To validate that NK cells could also delay rebound of an X4-tropic strain, we intravenously injected BLT mice with NL-HABC and then four weeks post-infection initiated ART (RAL/FTC/TDF) for six weeks to reduce active viral replication (Fig. 2b). Once viral loads were suppressed, ART was discontinued in the presence or absence of adoptively transferred allogeneic human peripheral blood NK cells similar to Fig. 1a. Six out of 14 mice (43%) in the group receiving NK cells rebounded and showed a significant delay in viral rebound compared to the control group, in which all 12 mice (100%) rebounded ($P = 0.0005$, Fig. 2c, d). Again, the frequency of human CD56$^+$CD3$^-$ cells in the blood peaked and then declined in the group receiving NK cells (Fig. 2c, right). Mice that did not rebound demonstrated a 4.8-fold higher frequency of human CD56$^+$CD3$^-$ cell in the blood compared to mice that did rebound ($P < 0.0001$, Fig. 2e), suggesting that NK cells were important to delaying rebound.

In addition, human immune cell engraftment between the mice that did or did not receive NK cells was not significantly different (Supplementary Fig. 5a–e). There was no significant difference between the pre-ART viral loads between mice that rebounded or did not rebound (Fig. 2f) or mice that did or did not receive NK cells (Fig. 3a). These results suggest pre-ART viral loads and the level of human immune cell engraftment likely did not contribute to differences in viral rebound. Among the mice that rebounded, there were no significant differences in the plasma viral loads at necropsy or level of cell-associated HIV RNA in the spleen and bone marrow between mice that did or did not receive NK cells (Fig. 3b, c), suggesting again that once rebound occurred viral replication was robust in these compartments.

Among the mice with rebound viremia, there were no significant differences in the level of cell-associated HIV DNA in the spleen and bone marrow at necropsy between the control and treatment groups (Fig. 3d), which was expected since rebound virus may reseed new target cells after ART interruption. Mice that did not display rebound viremia also had undetectable levels of cell-associated total HIV DNA in the spleens with the exception of one mouse in the NK group that had detectable HIV DNA in the spleen (mouse #2_9 in Supplementary Table 1). In addition, although all the control mice demonstrated rebound plasma viremia, we were unable to detect total HIV DNA from either the spleen in two out of twelve (17%) mice by PCR using 200 ng of input cell-associated DNA (mouse #2_3 and #2_4 in Supplementary Table 1), suggesting possible sampling limitation since only a small portion of the total spleen was analyzed for HIV DNA viral loads.

**NK cells reduce the viral growth after ART interruption.** In addition to measuring time to rebound after ART interruption, we compared viral growth kinetics between the mice that did or did not receive NK cells using the multiple timepoints in which viral loads were increasing prior to necropsy (Supplementary Fig. 6). We quantified the rate of virus spread following discontinuation of ART using a well-established mathematical model of virus dynamics (Methods)[47,48]. We focused on the exponential phase of virus growth during rebound viremia and fit this model to the experimental data and found that the rate of increase of virus spread in the control group upon rebound was $r = 0.181$ per day, while in the treatment group receiving NK cells it was $r = 0.087$ per day, approximately a 2.1-fold decrease in the viral spread rate[49]. These data suggest that NK cells target infected cells and thereby slow the rate of productive infection of new uninfected target cells in vivo. To further confirm these conclusions, we would have needed additional NK cell-treated

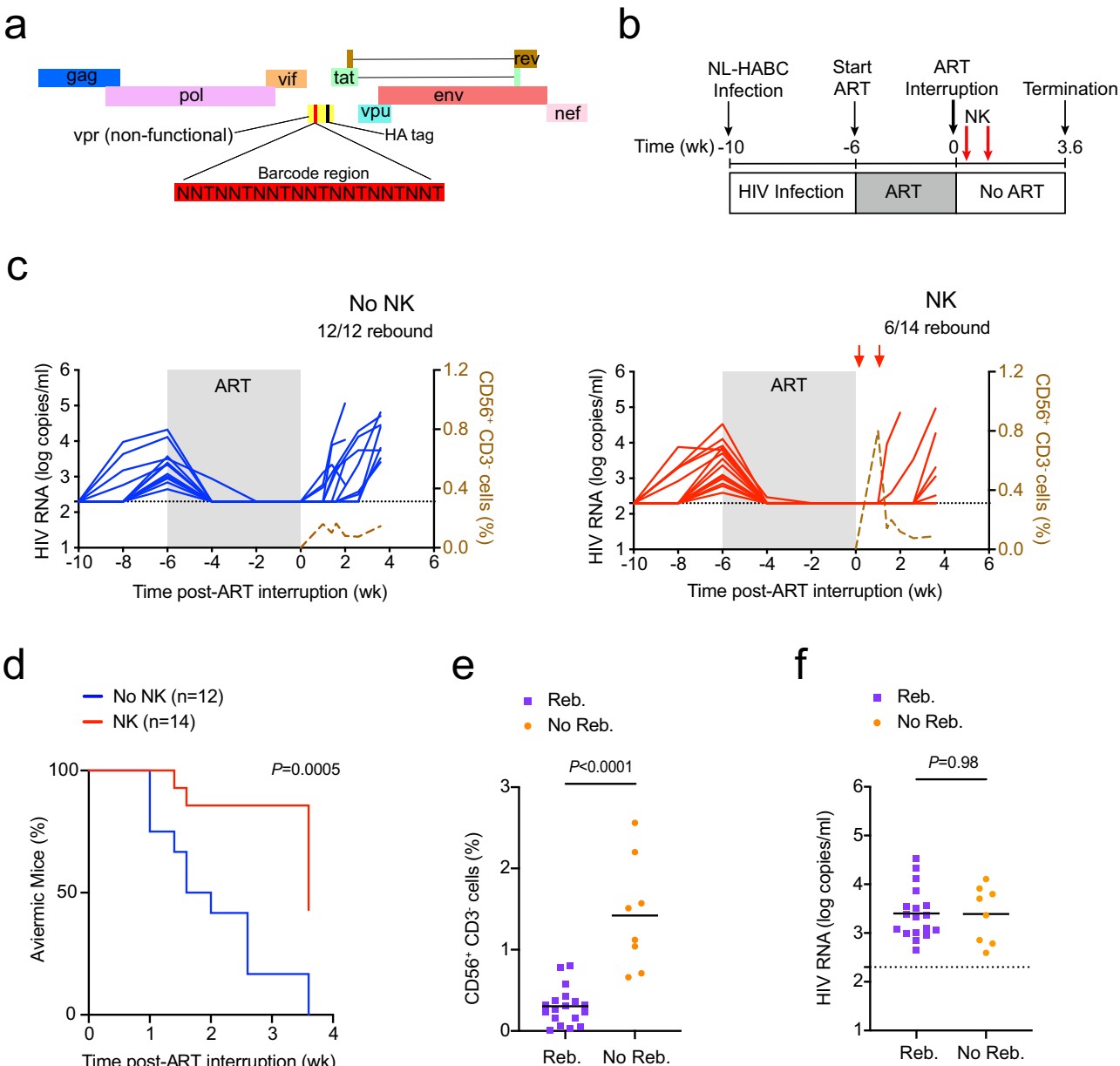

**Fig. 2 NK cells delay viral rebound after ART interruption. a** 21 bp barcode region (red) constrained by a thymine every third nt was inserted upstream of the hemagglutinin "HA" tag in a nonfunctional vpr of the HIV strain NL-HA[8]. **b** Schematic representation of experiment; acute infection of NSG-BLT with NL-HABC. **c** Longitudinal plasma viral loads for each infected animal at various timepoints in the No NK (blue) and NK (red) groups. Gray shading indicates ART treatment period of 6 weeks. A dashed brown line indicates the median frequency of CD56+CD3− cells detected in the blood. **d** Kaplan–Meier curves showing the frequency of aviremic mice after ART interruption. *P* value denotes statistically significant difference calculated by log-rank Mantel–Cox test. **e** Frequency of CD56+CD3− cells in the blood 5 days after the first injection of NK cells among mice that rebounded "Reb." (purple) or did not rebound "No Reb." (orange) after ART discontinuation. **f** Plasma viral loads for each animal before ART was initiated. *n* = 12 biologically independent animals in the No NK group, *n* = 14 biologically independent animals in the NK group (**c**, **d**). *n* = 18 biologically independent animals in Reb. group, *n* = 8 biologically independent animals in the No Reb. group (**e**, **f**). The black dotted line indicates the detection limit of 2.3 log RNA copies per ml (**c**, **f**). Horizontal bars represent the means (**e**, **f**). *P* values were calculated using a two-tailed Mann–Whitney test (**e**, **f**). Results are pooled from two independent experiments. Source data are provided as a Source Data file.

mice animals to have rebounded during the time frame of our experiments to track their viral growth curves.

**NK cells reduce the barcode diversity of rebounding virus.** In addition to characterizing the size of the reservoir by measuring the time to rebound after ART interruption, we utilized HIV barcoded technology to measure the viral clone diversity of the reservoir. We used an error-reduction deep sequencing strategy to

measure the number of HIV barcode variants that rebounded in the plasma, spleen, and bone marrow of each infected mouse after ART interruption. In this model, there were approximately 14,000 genetically different barcode clones in the virus preparation used to initially infect the animals (Supplementary Fig. 7a). Within the 21 bp barcode region, each barcode viral clone was different from one another on average by 10.5 bp, and the possibility that two barcodes were less than four bp different from each other was 0.04% (Supplementary Fig. 7b). Thus, barcode sequences with

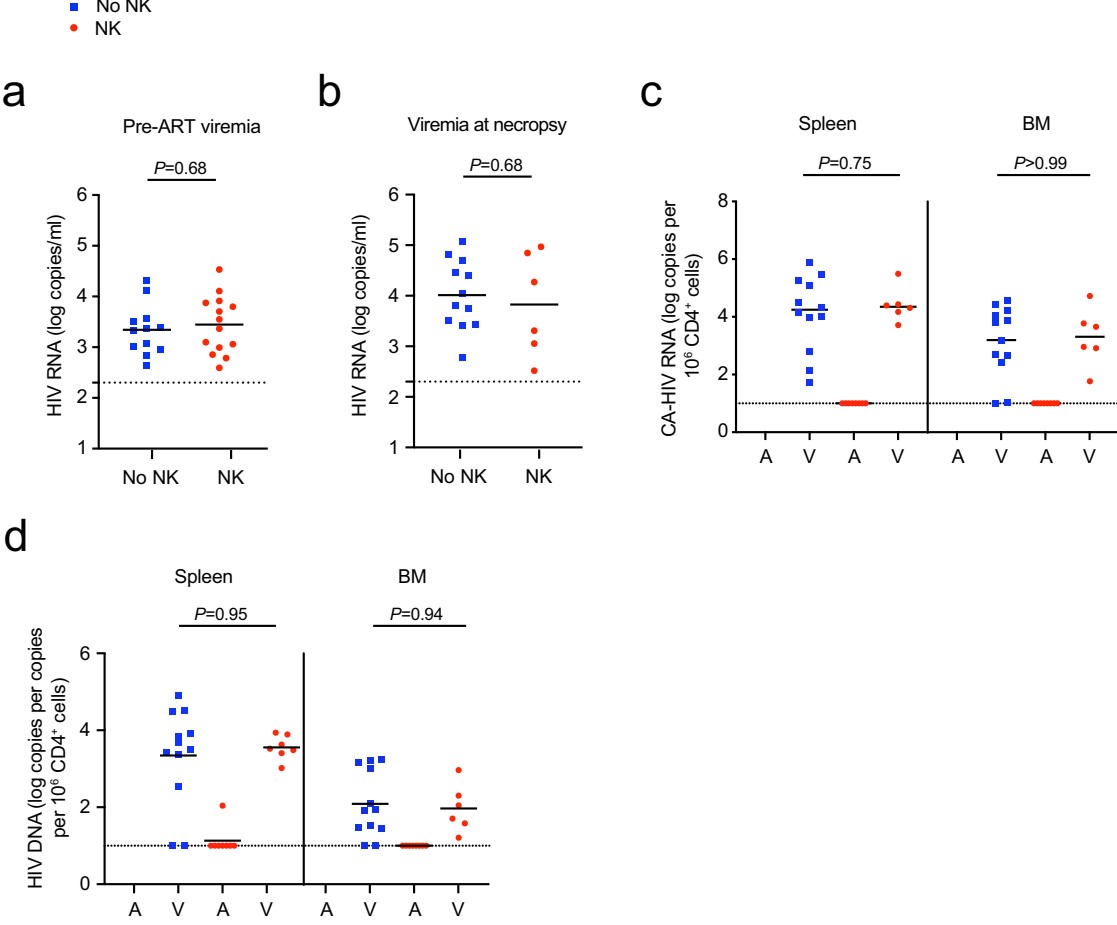

**Fig. 3 Viral RNA and DNA levels in NL-HABC-infected BLT mice treated with NK cells. a** Plasma viral RNA levels for each animal before ART was initiated in the No NK (blue) and NK (red) groups. **b** Plasma viral load at necropsy of the animals that rebounded after ART interruption. **c, d** Cell-associated HIV RNA (**c**) or DNA (**d**) is shown from the spleen and bone marrow "BM" of mice that were aviremic "A" or had rebound viremia "V" at necropsy. $n = 12$ biologically independent animals in the No NK group, $n = 14$ biologically independent animals in the NK group (**a**, **c**, **d**). Among the rebounding animals, $n = 12$ biologically independent animals in the No NK group, $n = 6$ biologically independent animals in the NK group (**b**). The black dotted line indicates the detection limit of 2.3 log RNA copies per ml (**a**, **b**) or 1.0 log RNA or DNA copies (**c**, **d**). Horizontal bars represent the means. $P$ values were calculated using the two-tailed Mann-Whitney test. Source data are provided as a Source Data file.

four or more bp differences from one another were considered unique clones. Although mutations in the barcode region may occur sporadically during viral replication, the chance of accruing at least 4 or more bp in the barcode region and thus mis-identifying a mutated barcode as an independent clone was likely negligible. We also employed a primer ID tag on each RNA molecule during cDNA synthesis to reduce sequencing errors and accurately quantify the starting number of RNA molecules (Supplementary Fig. 7c)[50,51]. We did not normalize the barcode number to viral load because the occurrence of each barcode is not uniform since a few clones of barcode virus usually domi-nated the viral population (Fig. 4a)[8]. To assess the sensitivity and reproducibility of our barcode quantification, we sampled RNA molecules from the same population twice and found the fre-quencies of barcodes with a three-log difference were significantly correlated between the two replicates (r = 0.92, $P = 6.7 \times 10^{-5}$, Supplementary Fig. 7d).

In contrast to total HIV RNA or DNA measurements, which are likely increased during ART interruption due to reseeding of the reservoir, we were able to utilize the overall diversity of circulating clones as an additional measure to study the effect that NK cells had on the viral reservoir. The number of unique barcodes rebounding in each mouse was 3.2-fold higher in the control group compared to the group receiving NK cells (4.7 vs

1.5 mean unique barcodes, $P = 0.0018$, Fig. 4b). Mice receiving NK cells also had significantly lower numbers of rebound barcode variants in the plasma, spleen, and bone marrow (Fig. 4c). The major rebounding barcode variants were distributed throughout the plasma, spleen, and bone marrow after ART discontinuation (Fig. 4a). We previously found that although 14,000 barcoded HIV clones were injected into humanized mice, on average only 20 to 50 barcodes were distributed in the mice prior to starting ART, and then after ART interruption, only one to 20 barcodes were detected during viral rebound[8]. Thus, there are likely in vivo bottlenecks that limit the number of barcode viral clones seeding the reservoir, and then further turnover of viral clones while animals are on ART. Overall, these results indicate NK cells eliminated some early rebounding viral clones, thereby decreasing the diversity of rebounding viral clones that were able to circulate during ART interruption.

Next, we performed an ex vivo co-stimulation of the splenocytes with anti-CD3 and anti-CD28 antibodies and then cocultured them with CEM cells to propagate viral outgrowth for 14 days. Although we were unable to detect total HIV DNA in the spleens and bone marrow of two control mice that had rebound viremia, we were able to induce replication-competent virus ex vivo measured by p24 ELISA from the splenocyte cocultures seeded with larger numbers of cells from these

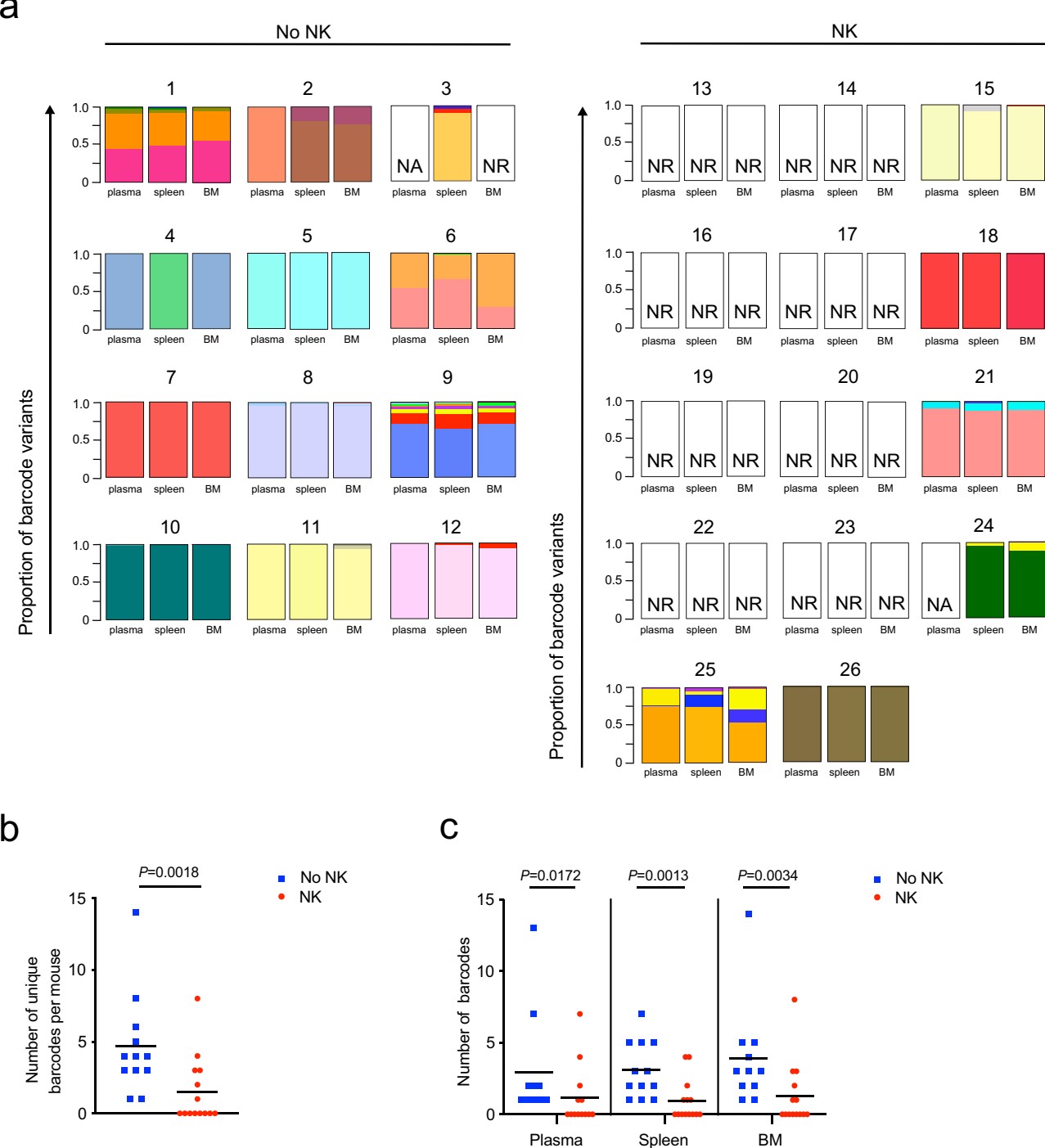

**Fig. 4 NK cells decrease barcode diversity in NL-HABC infected mice.** a Stacked bar plots display the relative frequencies and distribution of major barcode variants identified in the plasma, spleen, and bone marrow "BM" of mice treated with or without NK cells during ART interruption described in Fig. 2. Individual barcodes are represented by one color. "NR" is no rebound, undetectable HIV RNA copies measured by qRT-PCR. "NA" is no amplicon detected of the barcode region by nested PCR. b Number of unique barcodes quantified by deep sequencing of viral RNA from pooled organs per mouse in the No NK (blue) and NK (red) groups. $n = 12$ biologically independent animals in the No NK group, $n = 14$ biologically independent animals in NK group. c Number of unique barcodes from the plasma ($n = 11$ biologically independent animals in no NK group, 13 biologically independent animals in NK group), spleen, and BM ($n = 12$ biologically independent animals in no NK group, $n = 14$ biologically independent animals in NK group) of each mouse. Horizontal bars represent the means. P values were calculated using the two-tailed Mann-Whitney test. Source data are provided as a Source Data file.

animals (mice #2_3 and #2_4 in Supplementary Table 1). In addition, we were able to induce replication-competent virus from the splenocytes of seven out of eight (88%) mice that received NK cells even though they lacked rebound plasma viremia (mice #2_5, #2_6, #2_9, #3_10, #3_11, #3_13, and #3_14,

but not #2_8, induced virus in Supplementary Table 1). These results suggest that although NK cells can efficiently eliminate reactivated cells and their viral clones as they rebound, they were not effective at kicking and killing latently infected cells out of latency.

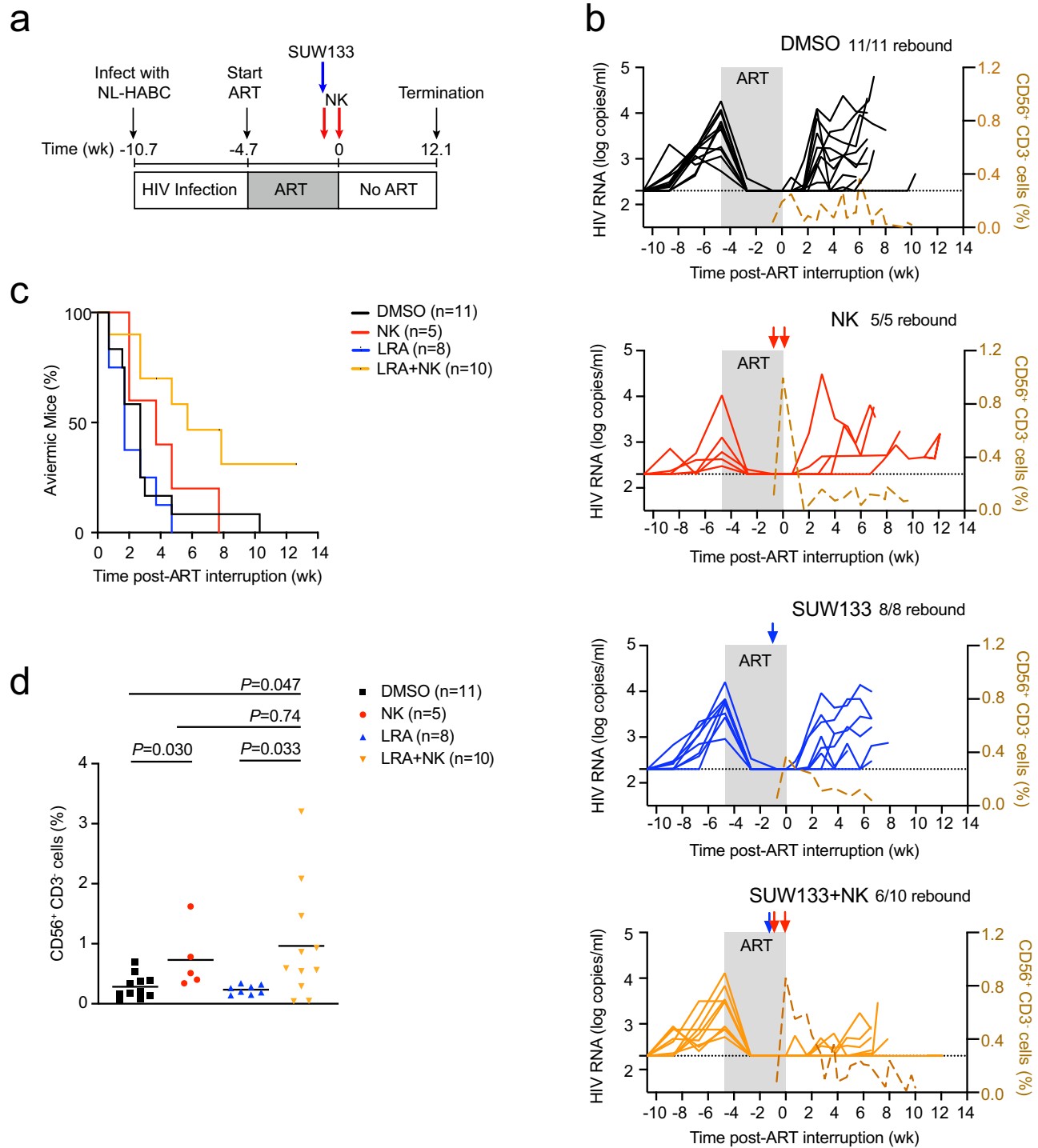

**Fig. 5 SUW133 plus NK cells delay viral rebound after ART interruption. a** Schematic representation of experiment; acute infection of TKO-BLT mice with NL-HABC. ART for 4.7 weeks. SUW133 (blue arrow) or control vehicle DMSO was administered 5 days prior to stopping ART. $5 \times 10^6$ allogeneic human peripheral blood NK cells (red arrows) were injected the same day as injection of SUW133 or DMSO and another homologous dose of cells were given 5 days later on the day of ART discontinuation. **b** Longitudinal plasma viral loads for each infected animal at various timepoints in the DMSO only (black), NK only (red), LRA only (blue), and LRA plus NK (orange) groups. Gray shading indicates an ART treatment period of 4.7 weeks. A dashed brown line indicates the median frequency of CD56+CD3− cells detected in the blood. The black dotted line indicates the detection limit of 2.3 log RNA copies per ml. **c** Kaplan–Meier curves showing the frequency of aviremic mice after ART interruption. **d** Frequency of CD56+CD3− cells in the blood 5 days after the first injection of NK cells. Horizontal bars represent the means. *P* values were calculated using the two-tailed Mann–Whitney test. $n = 11$ biologically independent animals in DMSO group, $n = 5$ biologically independent animals in NK group, $n = 8$ biologically independent animals in the LRA group, $n = 10$ biologically independent animals in LRA plus NK group (**b**–**d**). Results are pooled from two independent experiments. Source data are provided as a Source Data file.

**Kick and kill comprised of SUW133 and NK cells delays viral rebound**. We next hypothesized that combining a robust LRA with NK cells would effectively diminish the viral reservoir, as NK cells might eliminate cells induced to express virus due to the LRA activity. We selected SUW133[7], a synthetic bryostatin 1 analog, and PKC modulator, in which a single administration of this LRA activated CD4[+] T cells and efficiently reversed latency in vivo, but did not eliminate the viral reservoir on its own[6,52]. We infected BLT mice with NL-HABC, followed by ART to halt productive infection (Fig. 5a and Supplementary Table 2). Once mice were suppressed on ART, a subset of mice received a single intraperitoneal injection of 2 µg of SUW133 per mouse to reverse latency, followed by an intravenous injection of $5 \times 10^6$ allogeneic human NK cells 8 h later. We allowed five days for the reactivation and killing of activated infected cells to occur while the mice were maintained on ART. ART was then discontinued, and mice were concurrently administered one more intravenous injection of $5 \times 10^6$ NK cells. A subset of control mice received SUW133, but no NK cell injections. All remaining control mice received DMSO as a vehicle control instead of SUW133. A subset of the mice receiving DMSO also received NK cell injections. Plasma viral loads were monitored after ART interruption for up to 12.1 weeks (Fig. 5b). In these experiments, we turned to TKO-BLT mice, which can be robustly infected with HIV and are less susceptible to GVHD compared to NSG-BLT[53], which allowed us to monitor the mice after ART interruption for a longer interval.

Importantly, four out of ten (40%) mice receiving the combination of SUW133 plus NK cells displayed no rebound viremia despite monitoring animals for an extended time after ART interruption. In contrast, all control mice receiving SUW133 alone, NK cells alone, or DMSO alone rebounded during the lengthened interval of ART interruption (Fig. 5b). Mice receiving SUW133 plus NK cells demonstrated a significant or trend towards delay to rebound after ART interruption in comparison to the other groups ($P = 0.012$, SUW133 + NK vs DMSO; $P = 0.0037$, SUW133 plus NK vs SUW133; $P = 0.09$, SUW133 plus NK vs NK; log-rank Mantel–Cox test; Fig. 5c). The mice receiving SUW133 alone or NK cells alone did not demonstrate a significant delay in rebound compared to mice receiving DMSO alone ($P = 0.59$, SUW133 vs DMSO; $P = 0.47$, NK vs DMSO, log-rank Mantel–Cox test, Fig. 5c).

In contrast to mice that received NK cells in Fig. 2, in this experiment, all the mice receiving NK cells alone eventually rebounded, and no significant delay to viral rebound was observed compared to control mice receiving DMSO alone (Fig. 5c). To note, the mice receiving NK cells alone in this experiment initially appeared to have a delay in rebound compared to mice receiving DMSO alone during the first two to four weeks post-ART interruption, but as viral loads were monitored for a longer period after ART was discontinued this trend did not reach statistical significance. Importantly, in this experiment mice were administered NK cells while they were on ART, thus the level of peripheral blood CD56[+]CD3[−] cells peaked while animals were on ART (Fig. 5b), suggesting that HIV-expressing cells may not have been present for NK cells to target in the absence of an LRA.

To assess whether the SUW133 was activating latent cells in vivo, we measured the expression of the activation marker CD69 on CD4[+] T cells in vivo and found CD69 was significantly elevated on peripheral CD4[+] T cells in the blood five days after administration of SUW133 among mice that received either SUW133 plus NK cells or SUW133 alone (Supplementary Fig. 8a). There was no significant difference in CD69 expression on CD4[+] T cells between mice that received SUW133 alone or SUW133 plus NK cells. These results are consistent with the previous report that CD69 is maximally expressed on CD4[+]

T cells in vivo ~one to two days post-SUW133 injection, and then declined, but can be detected for up to nine days post-injection[6]. In addition, among the mice that received NK cells alone or in combination with SUW133, the level of CD56[+]CD3[−] cells in the blood increased over the five days post-NK cell injection as expected (Fig. 5b, d). Among the mice receiving SUW133 plus NK cells, the engraftment period of NK cells likely coincided with the period of latency reversal induced by SUW133, thereby allowing NK cell killing of latent cells upregulating activation or stress markers. Altogether, these results indicate that the combination of SUW133 and NK cells provided the most durable delay in rebound over these longer timescales compared with either individual treatment alone.

There was no significant difference between the pre-ART viral loads among the groups of mice (Supplementary Fig. 8b). Next, we assessed whether there was a correlation between time to rebound and pre-ART viral loads among the groups of mice. Pre-ART viral loads were inversely correlated with time to rebound in mice that received DMSO only ($r = -0.65$; $P = 0.030$, Pearson correlation test, Supplementary Fig. 8c), suggesting the extent of viral infection before ART initiation was important in determining the time to viral rebound after ART interruption. Similar trends were seen in mice that received SUW133 alone or NK cells alone, but not LRA plus NK cells. These results suggest that the effectiveness of the kick and kill approach may not significantly correlate to the extent of the initial seeding of the reservoir. In addition, the frequencies and absolute counts of human immune cells between the different groups of mice were not significantly different (Supplementary Fig. 9), suggesting levels of human immune cell engraftment likely did not contribute to differences in viral rebound between treatment groups.

Next, we analyzed plasma viremia and cell-associated HIV RNA in the tissues. Among the mice with rebound viremia, the viral loads at necropsy were not significantly different between treatment and control groups (Fig. 6a, b), which was expected since viral replication was allowed to proceed in rebounding animals. The viral RNA was deep sequenced to analyze the barcode diversity of the rebounding virus in each mouse. Importantly, mice receiving the combination of SUW133 plus NK cells had 1.1 mean unique barcodes per mouse, which was significantly lower than 4.7, 3.0, and 4.1 mean unique barcodes from mice receiving DMSO alone, NK cells alone, and SUW133 alone, respectively (Fig. 6c). There were no notable differences in the numbers of unique barcodes between the groups of mice receiving NK cells alone, SUW133 alone, and DMSO alone ($P = 0.25$, NK vs DMSO; $P = 0.52$, SUW133 vs DMSO; $P = 0.65$, NK vs SUW133; two-tailed Mann–Whitney test; Fig. 6c). A similar trend was also observed when comparing barcode diversities in the plasma, spleen, and bone marrow of these groups (Fig. 6d), indicating the combination of SUW133 plus NK cells more effectively diminished the diversity of rebounding viral clones compared to either SUW133 alone, NK cells alone, or the DMSO vehicle control. In addition, we found there was a significant inverse correlation between time to rebound and the number of unique barcodes among all the rebounding animals in this study (Fig. 6e), suggesting that the breadth of barcode diversity significantly correlates with the time to rebound after ART interruption. Thus, in addition to monitoring time to rebound after ART interruption, quantifying the breadth of unique barcodes may be an important measure to characterize the reservoir.

Among the four mice that received the combination of SUW133 plus NK cells and did not display rebound viremia, cell-associated HIV DNA was not detected in the tissues (Fig. 6f). In comparison, cell-associated HIV DNA was detectable from the spleen and bone marrow of all rebounding animals. Importantly,

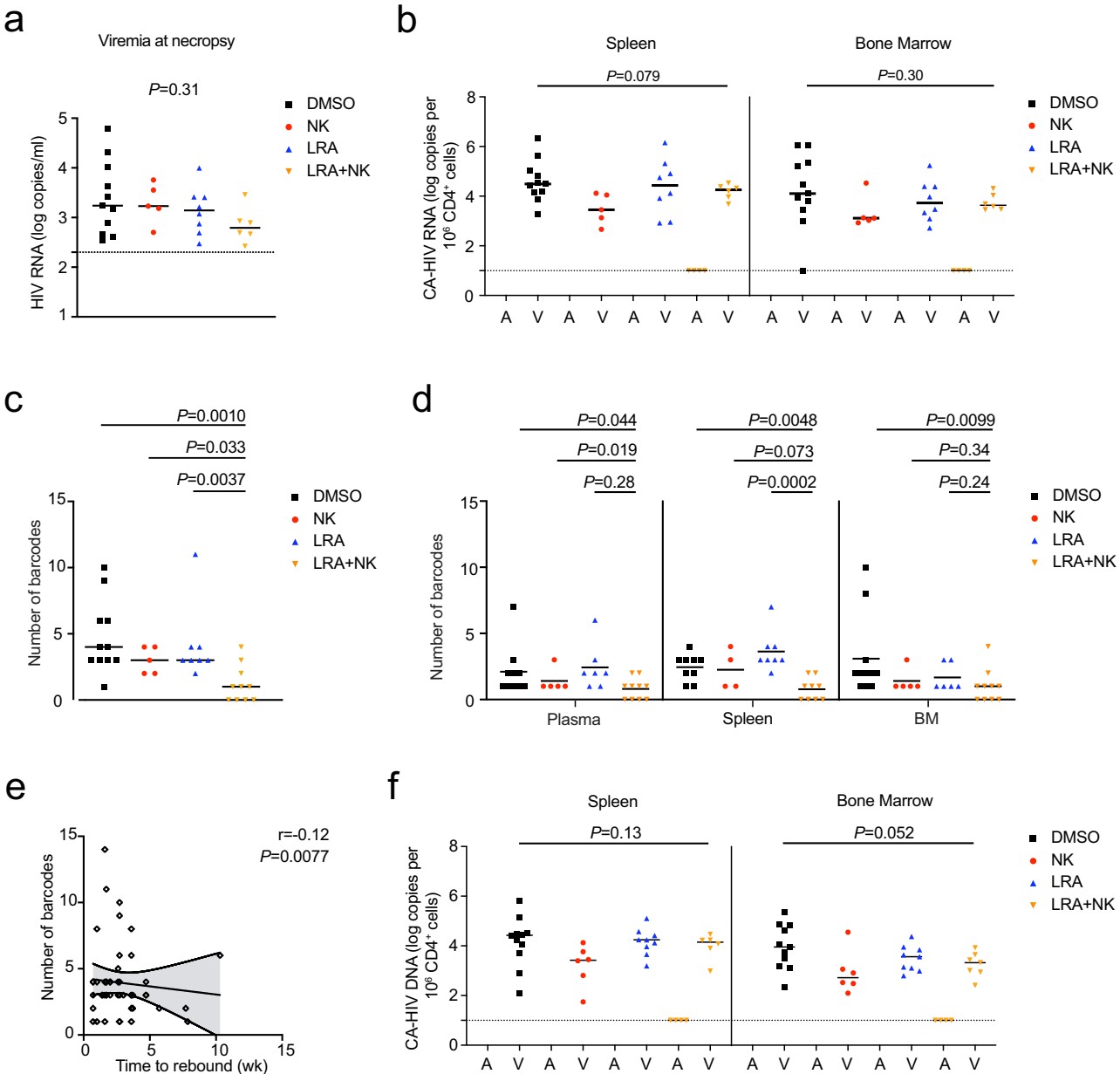

**Fig. 6 Levels of viral RNA, barcode diversity, and HIV DNA from infected mice treated with SUW133 and NK cells. a** Plasma viral loads of rebounding mice. *n* = 11 biologically independent animals in DMSO (black) group, *n* = 5 biologically independent animals in NK only (red) group, *n* = 8 biologically independent animals in LRA only (blue) group, *n* = 6 biologically independent animals in LRA plus NK (orange) group. **b** Cell-associated "CA" HIV RNA from the spleen and bone marrow "BM" of mice that were aviremic "A" or had rebound viremia "V" at necropsy. **c, d** Number of unique barcodes quantified by deep sequencing of viral RNA from pooled organs per mouse (**c**) or from the plasma, spleen, and BM of each mouse (**d**). **e** Scatterplot of a number of unique barcodes per mouse and time to rebound among all mice that rebounded after ART interruption. *n* = 48 biologically independent rebounding animals. Lines are linear predictions of time to rebound on the number of barcodes. The 95% confidence intervals of the fitted values are shown by gray areas. *r* Pearson correlation coefficient. **f** Total CA-HIV DNA from the spleen and BM of mice that were aviremic or had rebound viremia at necropsy. *n* = 11 biologically independent animals in DMSO only group, *n* = 5 biologically independent animals in NK only group, *n* = 8 biologically independent animals in LRA only group, *n* = 10 biologically independent animals in LRA plus NK group (**b–d, f**). Black dotted line indicates the detection limit of 2.3 log RNA copies per ml (**a**) or 1.0 log RNA or DNA copies (**b, f**). Horizontal bars represent the means (**a–d, f**). *P* values were calculated using one-way ANOVA Kruskal–Wallis test (**a, b, f**), two-tailed Mann–Whitney (**c, d**), or Pearson correlation test (**e**). Results are pooled from two independent experiments. Source data are provided as a Source Data file.

we obtained enough splenocytes from the four mice, which received SUW133 plus NK cells and lacked rebound viremia, to perform an ex vivo co-stimulation of the splenocytes. We were unable to detect replication-competent virus from the splenocyte cocultures by p24 ELISA (Supplementary Table 3). These results suggest that the administration of SUW133 plus NK cells

efficiently eliminated cells harboring replication-competent virus in the spleens in a subset of mice.

## Discussion

In this study, we demonstrate that the administration of allogeneic human peripheral blood NK cells delays viral rebound

after ART interruption in HIV-infected humanized mice. Importantly, we provide proof-of-concept demonstrating a kick and kill approach utilizing a single administration of the PKC modulator SUW133 as the kick and these NK cells as the killing agent delayed viral rebound after ART interruption, decreased the diversity of rebounding viral clones, and even eliminated infected cells harboring productive virus from the splenocytes of a subset of humanized mice infected with HIV-1.

Allogeneic NK cells have been widely and safely used in patients due to their strong alloreactivity toward leukemic cells[54]. In the field of infectious diseases, there are five clinical trials evaluating the use of allogeneic NK cells to treat COVID-19[55]. To date, allogeneic NK cells have not been used to clinically treat HIV infection. However, clinical studies have found that KIR and HLA mismatches between HIV-1 discordant sexual partners correlate with protection against HIV transmission[24]. In addition, allogeneic NK cells from healthy donors have been shown to inhibit HIV infection in vitro[24]. One preclinical study has shown that adoptively transferred allogeneic NK cells derived from human embryonic stem cells (hESC) inhibited acute HIV infection in humanized mice[25]. Here, we show the efficacy of allogeneic human peripheral blood NK cells against the HIV reservoir, especially when combined with the LRA SUW133 in a kick and kill approach.

The current findings are important because clinical trials employing kick and kill approaches have not previously yielded promising results[5]. However, recent kick and kill approaches have generated exciting findings in a few preclinical studies. For example, multiple LRAs combined with broadly neutralizing antibodies bNAbs was effective in delaying rebound after ART interruption in infected humanized mice[10]. TLR7 agonist plus Ad26/MVA vaccine[11] or bNAb PGT121[12] delayed viral rebound after ART interruption in rhesus monkeys infected with SIV or SHIV. Importantly, this study demonstrates a new and effective kick and kill combination utilizing a single administration of the LRA and PKC modulator SUW133 with allogeneic peripheral blood NK cells.

In this study, we employed the LRA SUW133 because we previously found a single administration activated CD4$^+$ T cells and reversed latency in vivo[8]. In addition, the now scaled GMP synthesis of bryostatin 1 can be readily applied to the more effective synthetic analog SUW133, thereby avoiding the time and cost-intensive large-scale harvesting of the marine source organism of bryostatin 1[56,57]. Consistent with our previous study[8], here we found mice treated with either SUW133 or SUW133 plus NK cells demonstrated a significant increase in CD69 expression on CD4$^+$ T cells in vivo (Supplementary Fig. 8a). In addition, we previously found SUW133 alone delayed rebound viremia for up to two to four weeks after ART interruption, but the splenocytes from the SUW133-treated mice generated replication-competent virus in ex vivo cultures, suggesting that all SUW133-treated mice would have eventually rebounded[8]. In the current study we monitored viral loads for a much longer period of ~12 weeks after ART interruption, which likely explains why we found all mice receiving SUW133 alone eventually rebounded. Another difference is the use of TKO-BLT mice in the current study, which have lower levels of endogenous immune activation compared with the NSG-BLT used in the previous study[8,58]. This diminished basal immune activation may make induction of high-level HIV expression in latently infected cells by an LRA alone more difficult to achieve, necessitating the use of a specific "kill" agent to eliminate these cells as opposed to viral cytopathic effects alone.

We suspect the timing of NK cellular injections is important in our kick and kill approach. The diversity of barcodes of rebounding viruses were reduced with NK cell treatment, which

suggests that NK cells inhibit or eliminate the reactivating latently infected T cells before viral propagation to new target cells can be completed. It suggests the window of activity of NK cells is after activation of T cells by LRA and before viral replication can occur. This could explain why NK cells were effective when administered after ART interruption (Fig. 2). In contrast, NK cells alone that were administered while virus replication was suppressed on ART failed to significantly delay viral rebound (Fig. 5c), presumably because NK cells do not act as a kick to increase HIV-expressing target cells in vivo. It is also possible that the diminished basal immune activation state of TKO-BLT mice[8,58] may have reduced the likelihood that NK cells would recognize any spontaneously activating latently infected cells on ART. Future studies to assess the optimization of treatment scheduling and lengthening of NK cellular engraftment may be helpful.

Others have shown NK cell-mediated killing of autologous HIV-infected target cells is partially dependent on interactions between the NKG2D receptor and one of its cell stress ligands ULBP2, which is upregulated by vpr in HIV-infected T cells[59,60]. We found that allogeneic NK cells effectively delayed viral rebound of NL-HABC, which does not express vpr (Fig. 2a). We suspect that multiple interactions involving inhibitory and activating receptors likely mediate alloreactive NK response to infected target cells in vivo. Other vpr-deficient HIV isolates were not evaluated in this study.

A limitation of our study is that due to the transient and low frequency of engrafted NK cells, we were unable to isolate NK cells ex vivo for additional studies. Another limitation of our study is that we did not use patient-derived HIV-1 isolates or R5-barcoded virus. However, we found allogeneic NK cells were effective at inhibiting the R5-tropic NFNSX in vitro and in vivo (Supplementary Fig. 2 and Fig. 1). We did not study the exact mechanism of alloreactive NK cell responses, which should be investigated in future studies. In addition, we were unable to humanely collect large longitudinal blood volumes from live animals to track viral barcode diversity in the plasma over time. Thus, barcode analysis was only performed on samples at necropsy. In addition, it is possible that our viral load assay may not have been able to detect small viral blips, intermittent viral replication, or a low level of productive infection. Lastly, although the humanized mouse model does not fully capture the immune response or viral latency in patients living with HIV, they provide an important preclinical model for HIV research.

In conclusion, this proof-of-concept study demonstrates that a kick and kill strategy comprised of LRA SUW133 plus allogeneic human peripheral blood NK cells substantially targets the HIV reservoir in an animal model compared to either the LRA or NK cells alone. Importantly, the utility of combining novel LRAs and NK cells opens a new paradigm for the HIV cure field.

## Methods

**Study design**. No statistical methods were used to predetermine sample size. The investigators were not blinded to allocation during experiments, so they could assess whether the treatments were being tolerated by the animals. However, initial data and sequence analysis was conducted in a blinded fashion with respect to treatment.

**Cells and cell lines**. De-identified PBMCs from healthy human donors were obtained under informed consent from the UCLA AIDS Institute Virology Core Laboratory under IRB approval then provided to investigators in an anonymized fashion. NK cells were isolated using CD56 MicroBeads (Miltenyi) with 75-90% purity (Supplementary Fig. 10a). Cells were stained with CD158(KIR)-FITC (clone HP-MA4), CD244(2B4)-PE (clone C1.7), NKp80-APC (clone 5D12), CD336(NKp44)-PerCp/Cy5.5 (clone P44-8), CD314(NKG2D)-PE-Dazzle594 (clone 1D11), CD159a(NKG2A)-PE/Cy7 (clone S19004C), CD56-Brilliant Violet421 (clone HCD56), CD3-Brilliant Violet 510 (clone Hit3a), CD337(NKp30)-Brilliant Violet605 (clone P30-15), CD335(NKp46)-Brilliant Violet650 (clone 9E2), CD57-Brilliant

Violet711 (clone QA17A04), and CD16-Brilliant Violet785 (clone B73.1) (all from Biolegend) and Ghost Dye Red 780 (Tonbo Biosciences) prior to analysis by flow cytometry for UMAP analysis. To isolate CD4$^+$ T cells, adherent macrophages were removed from PBMCs by culturing in flasks overnight in C10 (RPMI 1640 media supplemented with 10% vol/vol FBS (Omega Scientific), 1% L-glutamine, 1% penicillin/streptomycin (Invitrogen), 500 mM 2-mercaptoethanol (Sigma), 1 mM sodium pyruvate (Gibco), 0.1 mM MEM nonessential amino acids (Gibco), 10 mM HEPES (Gibco), and 20 ng per ml of recombinant human interleukin-2 (IL-2) (Peprotech). CD4$^+$ T cells were isolated using CD4 MicroBeads (Miltenyi). 293 T cell line was purchased from the American Type Culture Collection (ATCC) (ATCC CRL-11268). The following reagent was obtained through the NIH AIDS Reagent Program, Division of AIDS, NIAID, NIH: GHOST (3) CXCR4$^+$CCR5$^+$ cells from Dr. Vineet N. Kalamansi and Dr. Dan R. Littman (cat.# 3942).

**Transfections**. To generate HIV-1 virus supernatant, plasmids were transfected into 293 T cells in T-150 flasks, following the manufacturer's protocol for the BioT transfection reagent (Bioland). Virus-containing supernatant was harvested at day 2 post-transfection and passed through a 0.45 μm filter. Aliquots of virus were frozen at −80 °C and thawed immediately prior to use.

**NK and CD4$^+$ T cell cocultures in vitro**. CD4$^+$ T cells were cultured at $1 \times 10^6$ cells per ml in C10 media with 20 ng per ml of IL-2 and Gibco Dynabeads Human T-Activator CD3/CD28 (Thermofisher) per manufacturer protocol for 3 days. Then co-stimulated CD4$^+$ T cells were resuspended at $1.5 \times 10^6$ cells per ml in C10 media with 20 ng per ml of IL-2 and spin-infected with 800 ng of p24 NL4-3 or NFNSX per $1 \times 10^6$ cells at $1200 \times g$ for 2 h at 25 °C. The CD4$^+$ T cells were washed twice with media and cocultured with NK cells at an effector-to-target (E:T) ratio of 1:1 overnight at $1 \times 10^6$ cells per ml in C10 media with 20 ng per ml of IL-2. Flow cytometry was used to assess intracellular IFN-γ and CD107a levels at 24 h by staining with CD3-Pacific Blue (clone Hit3a), CD56-PE-Cy7 (clone HCD56), IFN-γ-PerCp (clone 4 S.B3) and CD107a-PE (clone H4A3) (all from Biolegend), and Ghost Dye Red 780 (Tonbo Biosciences). Infection was also assessed at 48 h by staining with p24 using FITC-conjugated antibody (clone KC57) (Beckman Coulter).

**Barcode analysis**. RNA was extracted from the barcoded virus supernatant using QIAamp Viral RNA Mini Kit (Qiagen) and from cells using RNeasy Mini Kit (Qiagen). cDNA was generated using SuperScript IV First-Strand Synthesis Kit (Invitrogen). The same amount of input RNA that was used for viral load measurement was also used for cDNA synthesis for barcode analysis. The barcode region was amplified by hemi-nested PCR using Phusion High-Fidelity DNA Polymerase (ThermoFisher Scientific). Each RNA molecule was tagged with a Primer ID[50,51]. Primer ID removal and cDNA purification was performed using an Purelink Quick PCR Purification Kit (Invitrogen). For the first PCR reaction, a fixed volume of 11 μl of cDNA was used along with the following primers: FW-5′-CTGACAGAGGACAGGTGGAACAAGC-3′ and BW-5′-GCCTTGCCAGCAC GC TCACAG-3′. The second PCR reaction used 2 μl of the first PCR reaction, the same reverse primer, and the following forward primer: FW-5′-AAGGGCCACA-GAGG GAGC-3′. The amplified fragment was ligated with the sequencing adapter, which had a six-nucleotide multiplexing ID to distinguish among different samples. Deep sequencing was performed with Illumina HiSeq3000 PE150. We ensured sequencing depth is tenfold higher than viral genome copies. Raw sequencing reads were de-multiplexed using the six-nucleotide ID. Sequencing error within the barcode region was corrected by filtering out low-quality reads (quality score <30) and unmatched base pairs between forward and reverse reads. We also used Primer IDs to correct sequencing errors. We found the frequency of Primer ID reads follows a bimodal distribution (Supplementary Fig. 7c). We filtered the Primer IDs using the frequency cutoff between the Poisson distribution of errors and normal distribution of real Primer IDs. For each Primer ID, the most frequently observed barcode was called. We then grouped the similar barcodes into clusters. Clustered barcodes represent sequences with ≥4 bp differences from one another (Supplementary Fig. 7b). Barcode clusters with less than 400 occurrences were filtered to remove handling and sequencing errors. To prove our barcodes quantification are an absolute number of clones, we sampled RNA molecules from the same population twice and found the number of barcodes is identical in the two samples (Supplementary Fig. 7d).

**In vitro HIV-1 infections**. GHOST (3) CXCR4$^+$CCR5$^+$ cells were cultured in DMEM containing 10% vol/vol FBS, 500 μg per mL G418 (Gibco), 1% penicillin/streptomycin, 100 μg per mL hygromycin (Sigma), and 1 μg per mL puromycin (Sigma). Cells were seeded into 24-well tissue culture plates at $5 \times 10^4$ cells per well. The next day media was replaced with various doses of HIV-1 and fresh media containing 10 μg per mL of polybrene (Sigma Aldrich). Plates were incubated for 2 h at 37 °C and then the media was replaced with 1 mL of fresh media without polybrene. Cells were incubated for a further 2 days, and then harvested by exposure to 0.05% trypsin (Gibco) in phosphate buffer saline (PBS) (Gibco) for 5 min, and then removed by agitation with media. Cells were collected and fixed in 3% paraformaldehyde then analyzed for GFP expression by flow cytometry (Supplementary Fig. 10b). CD4$^+$ T cells were isolated from PBMCs by

immunomagnetic selection (Miltenyi) following the manufacturer's instructions, then were co-stimulated with Dynabead CD3/CD28 human T-activator (ThermoFisher Scientific) per manufacturer's instructions and cultured in C10 media containing 20 ng per ml of IL-2 (Peprotech). For infection, $5 \times 10^5$ cells were exposed to HIV-1 in 200 μl of C10 media containing IL-2 and 10 μg per mL of polybrene. Cells were spin inoculated by centrifugation at $1200 \times g$ for 1.5 h at 22 °C. After spin-inoculation, cells were washed and resuspended in 200 μL of fresh C10 media containing IL-2. Viral infection was quantified by staining cells for p24 using PE-conjugated antibody to HIV core antigen (clone KC57) and analyzed by flow cytometry (Supplementary Fig. 10c). Cell-free supernatant samples were analyzed using HIV p24 enzyme-linked immunosorbent assay kit (Beckman Coulter).

**Mice**. All mice were maintained in the animal facility at UCLA. All experiments were performed in ethical compliance with the study protocol approved by the UCLA Animal Research Committee (ARC # 1996–058). Humanized bone marrow liver thymus (BLT) mice were constructed by the UCLA humanized mouse core using techniques described previously[8,61]. In brief, NOD.Cg-Prkdc$^{scid}$ Il2rg$^{tm1Wjl}$/SzJ or NSG mice or C57BL/6 Rag2$^{−/−}$γc$^{−/−}$CD47$^{−/−}$ or TKO mice[58] were obtained from Jackson Laboratories and bred at UCLA. Male and female mice were age-matched and between 6 and 8 weeks old were irradiated with 270 rads, and then pieces of fetal thymus and liver tissue were transplanted under the kidney capsule. Mice were then injected intravenously with $5 \times 10^4$ human fetal liver-derived CD34$^+$ cells isolated by immunomagnetic separation.

**Plasma viral load measurements**. For interval biweekly or weekly bleeds, 50 μl of blood was collected using EDTA-coated capillary tubes by retro-orbital bleed for viral load measurements. Whole blood was spun at $300 \times g$ for 5 min to separate plasma from the cellular fraction. Total RNA was extracted from plasma using QIAamp Viral RNA Mini Kit (Qiagen) per the manufacturer's protocol. HIV-1 RNA was quantified by qRT-PCR. The reaction mixture was prepared using Taqman Fast Virus 1-Step Mastermix (ThermoFisher Scientific) with 20 μl eluted RNA and a sequence-specific targeting a conserved region of the HIV-1 *gag* gene probe (FAM 5′-CCTTTTAGAGACATCAGAAGGCTGTAGACAAATACTGGG-3′). The forward and reverse primers sequences were FW-5′-GGGAGCTAGAAC GATTCGCAGTTAAT-3′ and BW-5′-ATAATGATCTAAGTTCTTCTGATCCTG TCTGAAGGGA-3′, respectively. Cycle threshold values were calibrated using standard samples with known amounts of absolute plasmid DNA copies. The quantitation limit was determined to be 200 copies per ml. At necropsy, more than 100 μl of blood was collected from each mouse, which allowed at least 50 μl of plasma to be analyzed for barcode analysis.

**Tissue harvest and processing**. Plasma was separated from blood as described above, and then the remaining layer was lysed using RBC Lysis Buffer (Biolegend) to obtain the PBMCs. Splenocytes were obtained by passing disaggregated tissue through a 40 μm filter. Bone marrow cells were obtained by grinding bones using a mortar and pestle. Blood, spleen, and bone marrow cells were then fixed for flow cytometry as described below. The remaining cell pellets were stored for DNA analysis or suspended in RLT buffer (Qiagen) for RNA analysis and frozen at −80 °C.

**Staining and analysis for flow cytometry**. Cells from animals were stained with fluorescently conjugated antibodies: CD69-Brilliant Violet 510 (clone FN50) or CD14-Brilliant Violet 510 (clone M5E2), CD3-Pacific Blue (clone Hit3a), CD8a-FITC (clone Hit8a), CD4-PE (clone RPA-T4), CD19-PE-Cy5 (clone SJ25C1), CD45-APC (clone 2D1), CD56-PE-Cy7 (clone MEM-188) (all from Biolegend) and Ghost Dye Red 780 (Tonbo Biosciences). All flow cytometry samples were run using a MACSQuant Analyzer 10 flow cytometer (Miltenyi) or Attune NxT. (Beckman Coulter). All data was analyzed using FlowJo v.10 (TreeStart, Inc) (Supplementary Fig. 10d).

**Cell-associated (CA)-HIV RNA and total HIV DNA**. CA-HIV RNA was extracted from lysed splenocytes and bone marrow cells using RNeasy Mini Kit (Qiagen). CA-HIV DNA was extracted from cell pellets using DNeasy Blood & Tissue Kits (Qiagen). Viral loads were measured by qPCR using 500 ng of CA-HIV RNA and DNA with the same primers as above. CA-HIV RNA and DNA are reported as the number of HIV-1 RNA or DNA copies per $10^6$ CD4$^+$ cells. The same amount of input RNA that was used for viral load measurement was used for Primer ID and barcode analysis. All cDNA was used as the template for the first round of PCR during the barcode analysis.

**Splenocyte coculture**. The spleens were aseptically collected from mice at the time of necropsy, passed through a 40 μm cell filter to achieve single-cell suspensions, and lysed using RBC Lysis Buffer (Biolegend). About $2.5 \times 10^6$ to $36 \times 10^6$ splenocytes were resuspended in C10 media containing Piperacillin/tazobactam (Wockhardt), 2.5 μg per mL Amphotericin B (Fisher), and 20 ng per ml recombinant human IL-2 (Peprotech) at a concentration of $5 \times 10^5$ to $2 \times 10^6$ cells per ml. The splenocytes were cocultured with $1 \times 10^6$ CEM cells and co-stimulated with 1 μg per ml anti-CD28 antibody (Tonbo biosciences; In Vivo Ready™ Anti-Human

CD28 (clone CD28.2) on 6-well tissue culture plates pre-coated with 50 μg/ml goat anti-mouse antibody (Invitrogen; Goat anti-Mouse IgG (H + L) Secondary Antibody) and 1 μg per ml of OKT3 antibody (Tonbo biosciences; Purified Anti-Human CD3 (clone OKT3))[4]. Twenty-four hours after co-stimulation, the culture media were doubled. After 3 days of activation, the cells were resuspended in fresh media at $5 \times 10^5$ cells per ml. For each subsequent passaging, the cells were cultured in a 17:3 ratio of fresh culture media and prior culture media at $5 \times 10^5$ cells per ml. The supernatants were sampled on days 7, 10, and 14 days post-co-stimulation and diluted in Triton X-100 (Sigma) in PBS at a final working concentration of 0.5% Triton X. The samples were sent to the CFAR Virology Core at UCLA for HIV p24 antigen ELISA.

**Mathematical modeling**. To quantify the rate of virus spread following cessation of ART, we utilized a model of virus dynamics[48,49] to track the temporal evolution of the free virus population, $V$, the infected cells population, $I$, and the susceptible target cell population, $S$. The model is given by the following set of ordinary differential equations (ODEs):

$$\frac{dS}{dt} = \lambda - dS - \beta SV; \quad (1)$$

$$\frac{dI}{dt} = \beta SV - aI; \quad (2)$$

$$\frac{dV}{dt} = kI - uV. \quad (3)$$

Here in Eqs. (1–3) $\lambda$ is the total rate of susceptible target cell production and $d$ is the (per cell) death rate of susceptible cells; $\beta$ is infectivity (per virus), $a$ is the per-cell death rate of infected cells, $k$ is the per-cell rate of virus production, and $u$ is the viral death rate; all rates measured in $(days)^{-1}$. The initial number of infected cells in the model corresponds to the infected cell population in which the virus has been activated following latency. Since we focus only on the exponential phase of virus growth during rebound, the susceptible target cell population can be assumed to be constant. If we further assume that the turnover of the virus population is significantly faster than that of the infected cells[49], the initial virus spread is described by the following ODE (4):

$$\frac{dI}{dt} = \beta' I - aI, \quad (4)$$

where $\beta' = \beta kS/u$ and the viral spread rate is given by $r = \beta' - a$.

The details of the fitting procedure and statistical analysis are as follows. To obtain the best fit for the viral spread rate, $r$, in the absence and in the presence of NK cell treatment, we used standard linear regression to fit the post-ART growth data across all experimental repeats for each of the two conditions (Supplementary Fig. 6), which yielded the estimate for the slopes. In order to evaluate whether the viral spread rate was different in the absence and in the presence of NK cells, we used z-statistics for the difference in regression slopes[62]. It was found that the viral spread rates are significantly different in mice that were and were not treated with NK cells ($p = 0.001$).

**Statistical analyses**. Statistical analyses were performed using FlowJo v.10, Graphpad Prism 8.4.3, and Python 2.7.

**Reporting summary**. Further information on research design is available in the Nature Research Reporting Summary linked to this article.

## Data availability

The data generated in this study are provided in the Supplementary Information/Source Data file. Source data are provided with this paper. The reads generated in this study have been deposited on SRA (Short Read Archive) database under the accession number (PRJNA694337). Source data are provided with this paper.

## Code availability

All codes are available on https://github.com/Tian-hao/BarcodeHIV, https://doi.org/10.5281/zenodo.5723824.

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

## Acknowledgements

We acknowledge support from the National Institutes of Health (AI155232 to J.T.K., K08CA235525 to C.S.S., AI131294 to J.A.Z. and M.D.M., AI124743 to J.A.Z. and P.A.W., and AI145038 to R.S.), the American Foundation for AIDS Research (110057-69-RGRL to J.A.Z.), the National Science Foundation (DMS1662146/1662096 to N.L.K. and D.W.), the National Center for Advancing Translational Sciences UCLA CTSI Grant (UL1TR001881 to J.T.K.), and the UCLA Center for AIDS Research (AI28697). The UCLA AIDS Institute and the McCarthy Family Foundation and UCLA Department of Medicine also provided support. ART for humanized mouse studies were kindly provided by Merck (RAL) and Gilead Sciences (TDF and FTC). Funding agencies did not play a specific role in the conception, design, data collection, analysis, the decision to publish, or preparation of this manuscript.

## Author contributions

J.T.K., M.D.M., J.A.Z. and P.A.W. designed the study and developed SUW133. J.T.K., C.C., B.L., C.S., K.F., H.C., M.K., M.D., M.S.A.S. and M.S. performed the immunological and virologic assays. J.T.K., C.S., M.D.M. and J.A.Z. analyzed/interpreted the data from immunological and virologic assays. R.S. conceived and supervised the barcode experiments. J.T.K, T.-H.Z., N.S., D.B., K.Y.R.B., Y.S., H.J., Y.D. and R.S. performed the barcode analysis. D.W. and N.L.K. led the mathematical modeling. J.T.K. and J.A.Z. wrote the manuscript with input of all co-authors.

## Competing interests

Stanford University has filed patent applications on SUW133 and related technology, which has been licensed by Neurotrope BioScience (Synaptogenix, Inc.) and Bryologyx Inc. P.A.W. is an adviser to both companies and a cofounder of the latter. J.A.Z. is a cofounder of CDR3 Therapeutics and is on the SAB of Bryologyx. The remaining authors declare no competing interests.
