## [Peer Review File · Nature Communications]

REVIEWERS' COMMENTS

Reviewer #1 (Remarks to the Author):

This manuscript entitled "Latency Reversal Plus Natural Killer Cells Diminish HIV Reservoir in vivo" compares the effect of allogenic NK cell transplantation with or without the latency reversal agent SUW113 on HIV rebound and HIV reservoir diversity in mice, following ART interruption. This work demonstrates that NK cells alone are capable of delaying and sometimes preventing viral rebound - for the first time - depending on the virus and mouse model used. Administration of a bryostatin analog, SUW113 immediately prior to ART interruption and NK transplantation further decreases rebound frequency and delays rebound when it occurs- pointing to a promising strategy to drive functional cure of HIV. Importantly, because these treatments also limit the diversity of the HIV reservoir - it is possible that adjunct interventions may help to further reduce the reservoir for full fledged functional cure. This study therefore provides a first conceptual picture of a therapeutic strategy to target the HIV reservoir, a task that has been nearly insurmountable. The study brings together novel utility of LRAs and NK cells, both tested individually, in a well controlled animal model, providing a new paradigm for the HIV cure field at large.

Comments:

1. The section titled "NK cells delay NFNSX rebound after ART interruption" has very similar results as later sections involving the barcoded HIV. Some clarity on how these sections build on one another would be useful to the reader otherwise they appear like very similar experiments/findings - or could perhaps be used as a simple validation?
2. In all cases the diversity of viral clones during rebound are an extremely small fraction of the approximate 14,000 barcoded HIV clones used during infection. Illustrating the diversity of circulating clones prior to ART would clarify that the limited clonal output is the result of sieving into/out of the reservoir as opposed to technical problems or issues with the replicative capacity of the barcoded clones.

3. In Figure 6d, the assignment of aviremic mice as having a diversity of zero is a little misleading or difficult to understand. They should be considered a different group as in Figure 6b and 6f and ultimately could be excluded from diversity comparisons.

Reviewer #3 (Remarks to the Author):

The authors should be commended for the thorough and deep revision made to their study and acknowledgement of the limitations raised by the review process. As written the manuscript is interesting evidence for “kick and kill” and a testament to the utility of humanized mice for such studies.

Minor point:

Supplemental Figure 4 legend refers to 239T cells instead of 293T cells.

Reviewer #4 (Remarks to the Author):

I am satisfied that all concerns have been addressed

We have enclosed a point-by-point response to the reviewer comments for manuscript NCOMMS-21-40900-T entitled, "Latency reversal plus NK cells diminish HIV reservoir in vivo" to Nature Communications. Our rebuttal responses are in blue below. We would like to thank the reviewers for all their comments.

REVIEWER COMMENTS

Reviewer #1 (Remarks to the Author):

This manuscript entitled "Latency Reversal Plus Natural Killer Cells Diminish HIV Reservoir in vivo" compares the effect of allogenic NK cell transplantation with or without the latency reversal agent SUW113 on HIV rebound and HIV reservoir diversity in mice, following ART interruption. This work demonstrates that NK cells alone are capable of delaying and sometimes preventing viral rebound - for the first time - depending on the virus and mouse model used. Administration of a bryostatin analog, SUW113 immediately prior to ART interruption and NK transplantation further decreases rebound frequency and delays rebound when it occurs- pointing to a promising strategy to drive functional cure of HIV. Importantly, because these treatments also limit the diversity of the HIV reservoir - it is possible that adjunct interventions may help to further reduce the reservoir for full fledged functional cure. This study therefore provides a first conceptual picture of a therapeutic strategy to target the HIV reservoir, a task that has been nearly insurmountable. The study brings together novel utility of LRAs and NK cells, both tested individually, in a well controlled animal model, providing a new paradigm for the HIV cure field at large.

Comments:

1. The section titled "NK cells delay NFNSX rebound after ART interruption" has very similar results as later sections involving the barcoded HIV. Some clarity on how these sections build on one another would be useful to the reader otherwise they appear like very similar experiments/findings - or could perhaps be used as a simple validation?

We have relabeled the section to "NK cells delay viral rebound of R5-tropic HIV after ART interruption." The next section is now entitled, "NK cells also delay rebound of X4-tropic barcoded HIV." We agree that there are similarities between the experiments using R5 and X4 HIV isolates. As reviewer 1 suggested we have inserted new text (line 173 of the clean version of the revised manuscript) that now highlights that the experiment detailed in the section, "NK cells also delay rebound of X4-tropic barcoded HIV" is a validation experiment, and also demonstrates reduction of numbers of individual activated HIV infected cells.

2. In all cases the diversity of viral clones during rebound are an extremely small fraction of the approximate 14,000 barcoded HIV clones used during infection. Illustrating the diversity of circulating clones prior to ART would clarify that the limited clonal output is the result of sieving into/out of the reservoir as opposed to technical problems or issues with the replicative capacity of the barcoded clones.

The small volumes of plasma and low cell numbers on longitudinal mouse bleeds prevents us from routinely tracking viral diversity prior to ART¹. However, we have previously found that although 14,000 barcoded HIV clones were injected, on average only 20-50 barcodes were found

distributed in the mice prior to starting ART. Thus, as Reviewer 1 suspected, there is an in vivo sieving or bottleneck that results in only a fraction of barcode viral clones seeding the reservoir, which is not due to technical problems with detecting the barcodes. We have now added this explanation (lines 246-510 of the clean version of the revised manuscript).

3. In Figure 6d, the assignment of aviremic mice as having a diversity of zero is a little misleading or difficult to understand. They should be considered a different group as in Figure 6b and 6f and ultimately could be excluded from diversity comparisons.

The aviremic, non-rebounding mice were included in this calculation. We used a high-sensitivity RT-PCR with a lower limit of detection of 10 copies, and these aviremic mice do indeed have no detectable barcodes present. In addition, the detection limit of our barcode protocol is as low as one barcode. Therefore, we believe it is an accurate representation of the data to include mice that did not rebound and with denoted zero barcodes. If we exclude the four LRA+NK mice that did not rebound, then there are only five mice that remain in the LRA+NK group. Even with only five mice in the LRA+NK group, we able to show that LRA+NK group have significantly lower number of barcodes than the DMSO control mice (1.5 vs 4 barcodes, $p=0.030$). Also, these five LRA+NK mice have lower number of barcodes compared to LRA only (1.5 vs 3 barcodes) and NK only mice (1.5 vs 3 barcodes), but these trends did not reach statistical significance ($p=0.0579$, $p=0.18$, respectively).

Reviewer #3 (Remarks to the Author): The authors should be commended for the thorough and deep revision made to their study and acknowledgement of the limitations raised by the review process. As written the manuscript is interesting evidence for “kick and kill” and a testament to the utility of humanized mice for such studies.

Minor point:

Supplemental Figure 4 legend refers to 239T cells instead of 293T cells.

We have corrected the error in Supplementary Figure 4 legend from 239T to 293T.

Reviewer #4 (Remarks to the Author):

I am satisfied that all concerns have been addressed

References

- 1 Marsden, M. D. *et al.* Tracking HIV Rebound following Latency Reversal Using Barcoded HIV. *Cell Rep Med* **1**, 100162, doi:10.1016/j.xcrm.2020.100162 (2020).